# Core outcome sets in symptomatic peripheral artery disease, COS-PAD: Study protocol for developing core outcome sets in symptomatic PAD utilising systematic reviews, interviews, and delphi consensus

Akam Shwan[1,2,3]*, Maria Gonzalez-Aguado[1], Rob D. Sayers[1,2,3], John S.M. Houghton[1,2,3]

1 Department of Cardiovascular Sciences, University of Leicester, Leicester, United Kingdom, 2 Leicester Vascular Institute, University Hospitals of Leicester NHS Trust, Leicester, United Kingdom, 3 National Institute for Health and Care Research Leicester Biomedical Research Centre – The Glenfield Hospital, Leicester, United Kingdom

* akam.shwan@nhs.net, as1610@leicester.ac.uk

## Abstract

### Background

Symptomatic peripheral artery disease (PAD) presents as intermittent claudication or chronic limb-threatening ischaemia (CLTI). PAD research suffers from wide heterogeneity and non-comparability of outcome measures. The solution is to develop a core outcome set (COS) – a minimum standard of outcomes developed and agreed by key stakeholders (patients, healthcare professionals, and researchers). There are currently no agreed COSs for research in PAD. The aim of this project is to develop two separate COSs for symptomatic PAD; for each of intermittent claudication and CLTI.

### Methods/Design

The COSs will be developed according to Core Outcome Measures in Effectiveness Trials (COMET) guidelines. Two comprehensive systematic reviews will be supplemented by qualitative interviews of patients and carers and focus groups of healthcare professionals and researchers. A three-round Delphi consensus process followed by a stakeholder meeting will agree the final COSs. Full ethical approval has been granted by Health Research Authority, HRA and Health and Care Research Wales, HCRW (Brighton and Sussex REC reference 24/LO/0258).

### Conclusion

Two separate core outcome sets for research involving patients with intermittent claudication and CLTI will be developed. This will aid study design and ensure

**Data availability statement:** No datasets were generated or analysed during the current study. All relevant data from this study will be made available upon study completion.

**Funding:** AS is fully funded as George Davies Research Fellow and RDS is partially funded as George Davies Chair of Vascular Surgery through George Davies Charitable Trust (Registered Charity Number: 1024818, England and Wales) The funding body did not have any role in the study design, data collection, analysis, decision to publish, or preparation of the manuscript.

**Competing interests:** RDS is the National Chair of the Vascular Clinical Reference Group and a National Specialty Advisor for vascular services – both roles for NHS England. The other authors declared no competing interests exist.

meaningful results of clinical trials to guide patient management and development of best-practice guidelines for symptomatic PAD.

## Study Registrations

COMET registrations: 1590 and 2650

## Background

### Peripheral Arterial Disease

Peripheral Artery Disease (PAD) is defined as progressive, flow limiting stenotic and/or occlusive disease of the peripheral arteries supplying the legs. [1] PAD is increasingly recognised as an important cause of cardiovascular morbidity and mortality that affects >230 million people worldwide. [1,2] Symptomatic PAD presents as two distinct subcategories, Intermittent Claudication and Chronic limb-threatening ischemia (CLTI). There are stark differences between the two subcategories in terms of epidemiology, demographics, symptomatology, management plans and outcomes. [2–4]

Intermittent claudication is pain, or fatigue in the lower extremities, caused by PAD, brought about on exertion and relieved by rest. Symptoms commonly occur in the calves but may also occur in the buttocks or thighs corresponding to the proximal level of arterial obstruction. [4] Intermittent claudication affects the quality of life (QoL) of patients; decreasing day to day activity, reduced productivity, difficulty of employment, and changing of daily life habits to alleviate symptoms with no direct risk to limb or life. [4,5] Chronic limb-threatening ischemia (CLTI) is the end stage of PAD defined by ischemic rest pain and/or tissue loss, and gangrene. [4] In patients with diabetes mellitus, diabetic foot ulceration, infection or gangrene may be the initial presenting symptoms instead of pain owing to the polyneuropathy secondary to diabetes altering sensation in the limbs and rendering the foot insensate to pain. [6] CLTI is the end stage of PAD leading to severely decreased QoL; recurrent hospitalisation, and imminent threat to limb and life with all-cause mortality of patients diagnosed with CLTI in 5 years approaching 50%. [6,7]

The management pathway of all patients with PAD includes risk modification with lifestyle changes (e.g., smoking cessation) and optimal medical therapy. Exercise, ideally in the form of a supervised programme, is first line therapy for those with intermittent claudication. Intervention may be offered to those with lifestyle limiting claudication in whom conservative approaches have failed to improve QoL. In CLTI, revascularisation is often necessary for limb salvage and/or even preventing death. [3. 5. 7]

### Core outcome set

In the context of clinical trials, an outcome is defined as a measurement or observation used to capture and assess the benefits (effectiveness) of a treatment or intervention or the risks (side effects) associated with the intervention being tested. [8] Clinical decisions in management of patients are made based on clinical research

and trial outcomes, thus the chosen outcomes need to be relevant to healthcare professionals, patients, and policy makers. Inadequacy and variability in the choice of outcome measures in research and clinical trials lead to research waste and result in incomparability of results from different studies of the same intervention. [8–10]

Core outcome sets (COS) are an agreed, minimum set of reported outcome measures for a specific condition. [8] The concept of a COS is to provide set of standard outcomes measures to be reported in clinical trials and studies of a particular condition. The standardised set of outcomes are regarded as the minimum required; further measures can be reported on the top of the core ones whenever deemed necessary by the researchers. There has been an increase in developing COS in various medical fields but to date, there are no COSs for PAD. [8,10,11]

### Rationale

It was agreed with the researchers and Patient and Public Involvement, PPI groups (people with PAD) that helped designing this study that the presentations and goals of therapy for people with intermittent claudication and CLTI are sufficiently different to require separate COSs. Therefore, the aim of this research project is to develop two separate COSs for use in clinical research of symptomatic PAD: one for intermittent claudication and one for CLTI despite the possibility of overlap to an extent between the COSs.

### Methods

This study will be undertaken in accordance with the COS development process described in The *Core Outcome Measures* in Effectiveness Trials (COMET) Handbook. [8] The approach is mixed-methods. The project is divided into four key successive steps: systematic review of current literature; semi-structured one to one qualitative interviews with patients and carers and focus groups with researchers and healthcare professionals; Delphi consensus process, and; a final stakeholder meeting to agree the final COSs (Fig 1).

### Step 1 systematic reviews

Two separate systematic reviews; one for studies including patients with intermittent claudication (The COMIC Study, COMET registration 1590) and the other including studies of patients with CLTI (COS-CLTI, COMET Registration: 2650, PRSOPERO: CRD42023412204) will be conducted. The purpose of the systematic reviews is to compile an exhaustive

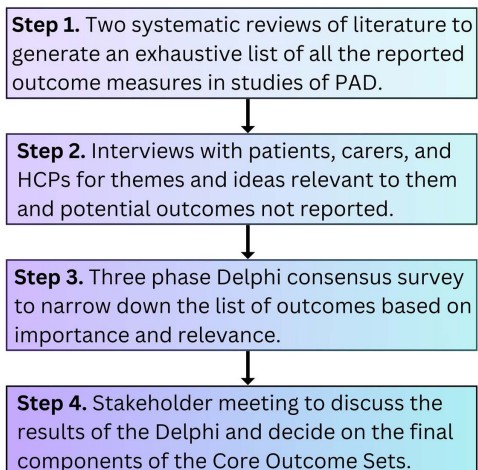

**Fig 1. Study flow diagram.**

list of all clinical, patient-reported, resource, composite, and surrogate outcomes in the literature involving patients with PAD. The comprehensive systematic reviews of the existing studies are carried out in accordance with the Preferred Reporting Items for Systematic Reviews and Meta-Analysis (PRISMA) Statement. [12]

**Study inclusion criteria.** Any clinical study reporting a minimum of one outcome involving patients with PAD will be included. The aim is to develop an "as saturated as possible" lists of reported outcomes, thus, all study types including observational and interventional studies will be included. Reference lists of any relevant systematic reviews found during screening will also be used to identify additional research studies not found in the original search.

**Search strategy.** Ovid generated electronic database search of Medical Literature Analysis and Retrieval System Online (MEDLINE), Excerpta Medica database (EMBASE), Cochrane Central Register of Controlled Trials (CENTRAL), and Cumulated Index to Nursing and Allied Health Literature (CINAHL) will be conducted. Medical Subject Headings (MeSH) and EMBASE Subject Headings (Emtree) as well as keywords with proximity and wild card operators for the conditions will be used. (Tables S1, S2 and S3).

The results of the search will be imported to Covidence (Covidence.org, Melbourne, Australia) for the purpose of study selection and duplication removal. Two independent reviewers will screen the study titles and abstracts. Full texts will then be screened by two reviewers independently to determine the list of included studies. Any conflict will be resolved through discussion.

**Data retrieval and synthesis.** Standardised proforma will be used for data extraction. Data extraction will include characteristics of the studies, participant demographics, outcomes, and outcome definitions as given by the original authors. Outcomes and their definitions will be extracted verbatim from the original manuscript. Risk of bias will not be assessed as the aim of the review is only to generate a list of outcomes from the studies not the efficacy of the outcomes. Again, data extraction will be carried out by two independent investigators, disagreements will be resolved through a senior investigator assessing the data independently. A narrative synthesis of results will be undertaken with the full list of outcome measures identified in the reviews tabulated and presented by the taxonomy described by Dodd et al. [11]

The systematic review of outcome measures used in studies of intermittent claudication has been completed and recently published. [13]

## Step 2 semi-structured interviews

Patients and their carers will be invited participate in a one to one semi-structured interview lasting approximately for one hour. (Table 1) The semi-structured interviews will flow along a specific topic guides codeveloped with the PPI group involved in designing the study. The interview questions will focus on participants' lived experiences, aspects of their diagnosis that are relevant to them, and potential outcomes that matter the most to them.

All participants will give signed, informed consent. To adapt to participants' preferences and individual circumstances, different interview formats; face to face or remotely via Microsoft Teams video call, or telephone calls will be offered. Interviews will take up to an hour.

Healthcare professionals and researchers will be invited to participate in focus groups (Tables 1 and 2) Additionally, one to one interview will be conducted for HCPs who are unable to attend a focus group. The focus groups will be conducted virtually on Microsoft teams and will last approximately 90–120 Minutes.

Face to face and telephone interviews will be recorded using encrypted, password locked voice recorders satisfying the institution and national data privacy policy. MS Teams interviews will be recorded within the app, which is approved by the sponsoring institution as well as the Health Research Authority (HRA) of the National Health Service (NHS).

**Data management and analysis.** Recordings will be transcribed by the primary researcher, coded and deidentified, and will be given a study ID number prior to analysis. Transcriptions will be uploaded to Nvivo 14 (Lumivero Inc, Massachusetts, USA) which will be used to compile codes and categories by the researcher and help manage the data.

**Table 1. Eligibility criteria of the participants.**

| Inclusion Criteria | Exclusion criteria |
|---|---|
| **Patients** | |
| • Patients with clinical diagnosis of symptomatic PAD (i.e., IC and CLTI).<br>• Asymptomatic patients who had any previous intervention for symptomatic PAD.<br>• Patients who have undergone amputation for complications of PAD. | • Asymptomatic individuals with PAD without prior intervention for PAD symptoms.<br>• Unable to read/speak/communicate in English.<br>• Lacking capacity to consent to participate.<br>• <18 years. |
| **Carers** | |
| • Adult carers of a patient participant in the study whom have consented for their carer/family member to participate in the study | • Unable to read/speak/communicate in English.<br>• Lacking capacity to consent to participate.<br>• <18 years |
| **Healthcare professionals and researchers** | |
| • Healthcare professional and researchers currently working with patients with PAD at any symptomatic stage of the disease. Professional defined as qualified professional registered with a professional regulatory body (e.g., GMC, NMC, HCPC registered) or non-qualified professional background (e.g., healthcare support worker). | • No experience of working with PAD patients.<br>• Unable to read/speak/communicate in English. |

IC, Intermittent Claudication; CLTI, Chronic Limb Threatening Ischaemia; PAD, Peripheral Artery Disease, GMC, General Medical Council; NMC: Nursing and Midwifery Council; HCPC, Health and Care Professions Council.

**Table 2. Roles and areas of expertise of the professional Stakeholder Meeting and Focus Groups invitees.**

| Roles and Expertise |
|---|
| • Vascular and Endovascular Surgeon<br>• Interventional Radiologist<br>• Vascular Anaesthetist<br>• Geriatrician<br>• Interventional Cardiologist<br>• Specialist Vascular Nurse<br>• Registered Nurse/Community Nurse<br>• Exercise Medicine Specialist<br>• Physiotherapist<br>• Occupational Therapist<br>• Rehabilitation specialist/nurse<br>• General Practitioner<br>• Vascular Scientist (Vascular Technician)<br>• Podiatrist<br>• Orthotist/Prosthetist<br>• Industry Representative |

Data will be analysed through Reflexive Thematic Analysis (RTA), which is an iterative process, aligning with phenomenology. [14] There will be continual review of the data to produce themes and ideas. The produced themes will serve as blueprint for the outcomes generated directly as well as indirectly from the interviews.

To enhance the rigour of the research, all the qualitative interviews and focus groups will be conducted as per the Consolidated Criteria for Reporting Qualitative Research, COREQ checklist. [15]

**Sampling and sample size.** Due to the qualitative nature of this part of the study, data will be collected until thematic saturation has been reached, defined as no emergence of new themes during the interviews. We are initially aiming to recruit fifteen patients and five carers for each of intermittent claudication and CLTI. Three focus groups (plus an additional control meeting if needed) will be conducted. Approximately six to ten participants will participate in each focus group with a minimum of three professional groups represented at each meeting (Table 2). If required,

additional one to one semi-structured interviews will be arranged with HCPs from roles not represented in any of the focus groups.

Maximum variation purposive sampling method based on age, sex, ethnicity and time since diagnosis for patients and role and seniority for HCPs will be completed to enable a wide variation of opinions and views to be obtained within the sample. This is supported by the underpinning theory of phenomenology and reflexive thematic analysis method. Should there be difficulty with recruiting with maximum variation purposive sampling method, the researchers will employ alternative sampling methods such as a convenience or snowball sampling strategy.

**Recruitment.** Clinicians working with PAD patients at the Leicester Vascular Institute, UHL NHS Trust, will identify patients admitted to the ward or during their routine clinic appointment, for their possible eligibility. Patient and carer participants may also self-identify. Leaflets will be circulated to various charities to advertise the research study with patients and cares as well as online posts and posters through charities and patients' groups via our PPI group.

Potential healthcare professionals and researchers working with vascular patients will be identified through other HCPs, regional, national, and international groups and societies for HCPs who look after PAD patients. Additionally, HCPs may get in contact with the research team through advertising via email networks, posters, social media, or through professional networks. Recruitment is expected to start from the day of ethical approval until the start of the Delphi Consensus surveys.

## Step 3 delphi consensus surveys

A longlist of outcome measures will be generated from the systematic reviews and qualitative interviews and focus groups to be used in a consensus process to identify the outcomes of greatest importance to patients, carers, healthcare professionals, and researchers. Given the expected volume of the outcomes, national and international experts from different specialities and disciplines in managing patients with PAD, researchers, academics as well patients and carer representatives will be invited to hold an initial stakeholder meeting (Table 2). The panel will narrow down the longlist of outcomes to carry over to the Delphi process.

A two to three round consensus survey will be conducted in accordance with the Delphi consensus process. [16] An e-Delphi methodology will be used as it facilitates participation on a global level. Paper based surveys will be sent out to any participant who will not be able to complete the online survey (on request). The e-Delphi survey will be conducted online using Jisc Online Surveys (Jisc, Bristol, UK).

In each round of the e-Delphi survey, participants will be prompted to grade each outcome item using a 9-point Likert-like scale based on the importance of each item on a scale from 1 (not important) to 9 (critically important). The threshold for consensus will be defined at ≥70%.

During the first round, outcomes which achieve ≥70% ratings between 7–9 will be deemed to be essential for inclusion and will be put forward for final COS consideration. Outcomes which achieve ≥70% ratings of 1–3 will be determined as not important and therefore excluded. Outcomes which do not reach the threshold of consensus will be put forward for round two of the e-Delphi. In addition to rating outcomes, participants will again be asked in a free-text format to suggest any other outcomes they find relevant as well as any further commentary.

In the second round of e-Delphi, the questionnaire will compose of non-consensus items and any new items suggested in round one. Next to each item, participants will be reminded of what rating they gave in the previous round and the mean scores given by each stakeholder group in the previous round will be displayed for each item. In this way, participants will be able to revise their initial score with the additional knowledge of other participants' responses. Again, any outcome culminating in more than a 70% rating of 7–9 will be added to the final COS consideration, and negative consensus items will be excluded. Round three if needed, will be a similar process for the non-consensus items from round two. Any no consensus item will be carried over to the final stakeholder meeting for consideration of either inclusion or exclusion.

**Recruitment.** Similar method of patient, carer, HCP, and researcher identification to the qualitative phase of the study will be utilised. Participants of the qualitative interviews can directly consent to participate in the Delphi surveys during the consent process for the interviews. As it is a survey-based exercise, exact number of the participants are difficult to determine but ideally the more the better, a uniformly distributed, diverse target of at least 100 participants is deemed satisfactory. All efforts will be made (including purposive sampling) to ensure broad representation from patients and carers as well as all key stakeholder professional groups on a global level.

A specific PIS for the Delphi process along with privacy notice and consent for participation will be provided to the invited participants at the time the first phase of the Delphi Consensus. Printouts of the PIS, Privacy notice and consent will be included along the survey for the participants opting for paper-based survey. Recruitment is expected to start from 1st May 2025–31st October 2025.

## Step 4 COS consensus meeting

Following the completion of the Delphi consensus process, the panellists of the initial stakeholder meeting will be invited to attend a final meeting to discuss the results of the consensus survey and agree on the final core outcome sets. As per the recommendation of the COMET Handbook, the Nominal Group Technique, a highly-structured group interaction framework, will be utilised to aid this process. [17]

## Dissemination plan

For the Core Outcome Sets to be effective and utilised in future trials and studies, the dissemination plan is to present the results at international conferences and scientific meetings as well as publication in high impact journals.

Furthermore, lay summaries will be circulated across social media platforms and study participants. Formal statement letters will be sent out to all relevant professional societies, charities, and patient groups informing them about the key details of the core outcome sets and the importance of their utilisation in future research. The PPI group will help inform the dissemination strategy.

## Ethics approval

Full ethical approval has been granted by Health Research Authority, HRA and Health and Care Research Wales, HCRW (Brighton and Sussex REC reference 24/LO/0258).

## Patient and public involvement

A PPI group involving patients and carers, with experience of being treated at the Leicester Vascular Institute has been established for this study. Members of the PPI group have been directly involved in the development of the study, acceptability of the research, the information material and provided insight on the burden of the study from a patient's perspective. The group will continue to meet throughout the duration of the study to be informed of the progress and advise on the ongoing delivery of the study and dissemination of the final COS.

## Discussion

The current PAD literature suffers from heterogeneity, non-comparability, and less focus on the most relevant outcomes to patients, all together leading to significant amount of research waste. [9] Previous attempts at grouping of outcomes in PAD, such as the outcomes recommended in the Global Vascular Guidelines on Management of CLTI and International Consortium of Vascular Registries Consensus Recommendations for Peripheral Revascularisation Registry Data Collection are important steps to improve reporting in trials involving patients with PAD. [18,19] However, their use are limited to specific scenarios, and above all patients did not have direct input in designing the grouping standards. Patients and

carers are central to developing core outcome set methodology described and defined by the COMET initiative to ensure that they are meaningful to those living with the disease. [8]

This project will develop two separate core outcome sets for the two subcategories of symptomatic PAD; intermittent claudication and CLTI. This is crucial to improve the quality, comparability and relevance of future PAD research. The decision of developing two separate COS for intermittent claudication and CLTI instead of one COS for PAD was heavily influenced by PPI input in the initial stages of the project. The experiences of those with intermittent claudication and CLTI are quite different necessitating for different set of outcomes to capture the most relevant endpoints for each group of patients. This approach is backed by up by research from Kengne and international guidelines make separate recommendation for intermittent claudication and CLTI. [18,20,21]

However, developing core outcome sets is only the first step, adherence to the use of the COS is equally important. As reported by the Zywicka *et al*, there are extremely poor adherence to available methodological standards in RCTs investigating endovascular interventions in PAD. [22] This can be inherently true for observational studies as well. In order to ensure the best chances of implementation, wide dissemination to all relevant stakeholders is essential. While this study will the define the COS, it will not determine the best way to measure each outcome. Wide variations in definitions of reported outcome and lack of specific tools, especially PROMS are concerning. [23] The COS may include important outcomes that have not got agreed way of measuring yet. Further work will likely be needed afterwards to determine the best ways to measure each item in the COS.

## Conclusion

Although there is a substantial amount of high-quality research evaluating the efficacy of management of patients with PAD, current work is limited by the quantity and heterogeneity of reported outcome measures which limits comparison and pooling of available data. Development of core outcome sets for research in PAD is essential for capturing the most important issues facing patients, carers, healthcare professionals, researchers as well as policy makers in order to generate high quality and robust evidence in management of patients with symptomatic PAD.

### Open access

For the purpose of open access, the author has applied a Creative Commons Attribution license (CC BY) to any Author Accepted Manuscript version arising from this submission.

### Supporting information

**S1 Table.  Search strategy of the Reported Outcomes in Studies of Intermittent Claudication Systematic Review, conducted in Medline via Ovid.**
(DOCX)

**S2 Table.  Search strategy of the Reported Outcomes in Studies of Intermittent Claudication Systematic Review, conducted in Embase via Ovid.**
(DOCX)

**S3 Table.  Search strategy of the Reported Outcomes in Studies of Chronic Limb Threatening Ischaemia Systematic Review.**
(DOCX)

### Author contributions

**Conceptualization:** Akam Shwan, Rob D. Sayers, John S.M. Houghton.

**Data curation:** Akam Shwan, John S.M. Houghton.

 

**Formal analysis:** Akam Shwan, Maria Gonzalez-Aguado, Rob D. Sayers, John S.M. Houghton.

**Funding acquisition:** Rob D. Sayers.

**Investigation:** Akam Shwan, John S.M. Houghton.

**Methodology:** Akam Shwan, Maria Gonzalez-Aguado, Rob D. Sayers, John S.M. Houghton.

**Project administration:** Rob D. Sayers, John S.M. Houghton.

**Resources:** Rob D. Sayers.

**Software:** Rob D. Sayers.

**Supervision:** Maria Gonzalez-Aguado, Rob D. Sayers, John S.M. Houghton.

**Validation:** Maria Gonzalez-Aguado, Rob D. Sayers, John S.M. Houghton.

**Writing – original draft:** Akam Shwan.

**Writing – review & editing:** Maria Gonzalez-Aguado, Rob D. Sayers, John S.M. Houghton.

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
