## [Decision Letter · Decision Letter 0]

Dear Dr. Shwan,

Thank you for submitting your manuscript to PLOS ONE. After careful consideration, we feel that it has merit but does not fully meet PLOS ONE’s publication criteria as it currently stands. Therefore, we invite you to submit a revised version of the manuscript that addresses the points raised during the review process.

We look forward to receiving your revised manuscript.

Kind regards,

Biswabandhu Jana, Phd

Academic Editor

PLOS ONE

Journal Requirements:

4.  We notice that your supplementary tables are included in the manuscript file. Please remove them and upload them with the file type 'Supporting Information'. Please ensure that each Supporting Information file has a legend listed in the manuscript after the references list.

Reviewers' comments:

Reviewer's Responses to Questions

**Comments to the Author**

1. Does the manuscript provide a valid rationale for the proposed study, with clearly identified and justified research questions?

Reviewer #1: Yes

Reviewer #2: Yes

2. Is the protocol technically sound and planned in a manner that will lead to a meaningful outcome and allow testing the stated hypotheses?

Reviewer #1: Yes

Reviewer #2: Yes

3. Is the methodology feasible and described in sufficient detail to allow the work to be replicable?

Reviewer #1: Yes

Reviewer #2: Yes

4. Have the authors described where all data underlying the findings will be made available when the study is complete?

Reviewer #1: No

Reviewer #2: Yes

5. Is the manuscript presented in an intelligible fashion and written in standard English?

Reviewer #1: No

Reviewer #2: Yes

You may also provide optional suggestions and comments to authors that they might find helpful in planning their study.

Reviewer #1: This type of work has already been published.

1. There is lack of Novelty in this study as similar COS for PAD is likely to have been developed.

2. There may be difficulty in achieving consensus among stakeholders through Delphi method.

3. The paper doesnot mention strategy to mitigate Biasness and surety of having proper representation.

4. There is no mention of preliminary finding related to this work

5. Although the manuscript is thorough, certain sections , like introduction and discussion, could be more streamlined for clarity. Narrative flow could be enhanced by concise phrasing and reducing the redundancy.

6. There is sentence error in some section of this paper.

Reviewer #2: This study is a well-written and methodologically sound study protocol addressing an important gap in peripheral artery disease (PAD) research—the lack of standardized core outcome sets (COSs). The authors have followed recognized methodological standards (COMET, PRISMA, COREQ) and outlined a comprehensive, mixed-methods approach including systematic reviews, qualitative interviews, Delphi surveys, and a final consensus meeting. The engagement with Patient and Public Involvement (PPI) is commendable and strengthens the relevance of the outcomes.

However, there are a few areas where clarifications, minor improvements, and additional details would strengthen the manuscript further.

• Although the justification for two separate COSs (for intermittent claudication and CLTI) is clear, the potential for overlap between the two outcome sets (e.g., QoL measures) should be discussed in more depth.

• While the Delphi process is well described, it is unclear how "borderline" outcomes (those close to the 70% threshold) will be handled.

• The target of 100 participants for the Delphi survey is appropriate, but greater emphasis on ensuring diversity across geographical regions (especially since PAD presentation can vary) would strengthen the plan.

• While the data availability statement is compliant, given that this is a COS development, transparency about access to the list of outcomes generated (even preliminary) would be important.

• Occasional minor typos, a final proofread before acceptance is recommended.

• Figure 1 (Key Steps of the Study) is mentioned but not included in this file.

• The search strategies are well presented; however, the timeframes for searches (especially if new studies are published during the study period) should be clarified.

**Do you want your identity to be public for this peer review?** For information about this choice, including consent withdrawal, please see our Privacy Policy

Reviewer #1: No

Reviewer #2: **Yes: ** Venkatesh Pooladanda

---

## [Author Response · Author response to Decision Letter 1]

20 Jun 2025

Dear reviewers,

We are really grateful for taking time to thoroughly review this manuscript and suggest revisions and amendments through your insightful comments which we are sure have made the manuscript much better and understandable.

We have addressed all your comments and made changes and amendments in the respective sections of the manuscript.

Thank you again, your feedback is much appreciated.

Reviewer 1

1. There is lack of Novelty in this study as similar COS for PAD is likely to have been developed.

There are core outcome sets developed for various conditions in the field of vascular surgery such as abdominal aortic aneurysm by Machin et al, major lower limb amputations by Ambler et al, and intervention for diabetic foot related ulceration by Staniszewska et al but to our knowledge is no any published core outcome set in symptomatic peripheral artery disease. Neither any COS is under development for each subcategory of PAD.

2. There may be difficulty in achieving consensus among stakeholders through Delphi method.

Thank you so much for the feedback. Members of the stakeholder committee will not participate in the Delphi consensus surveys. The Delphi surveys will be sent out to all eligible patients, carers, and healthcare professionals globally to aid with consensus on the longlist of the outcomes generated from the systematic reviews and qualitative interviews finetuned by the members of the stakeholder committee. At the end of the Delphi process, the stakeholder committee will meet again to ratify the results of the Delphi survey. Nominal Group Technique, NGT is used to achieve consensus during the stakeholders meeting. Further details regarding the consensus process are updated in the revised protocol.

3. The paper doesn’t mention strategy to mitigate Biasness and surety of having proper representation.

4. There is no mention of preliminary finding related to this work

Thank you for highlighting this but we have indeed mentioned and cited the published work prior to this submission. Kindly refer to the Reference 13, directly citing the published first systematic review i.e., reported outcomes in studies of intermittent claudication. As the CLTI systematic reviews and papers from the qualitative phase are yet not published hence cannot be referred to directly.

5. Although the manuscript is thorough, certain sections, like introduction and discussion, could be more streamlined for clarity. Narrative flow could be enhanced by concise phrasing and reducing the redundancy.

Thank you so much for the suggestion and recommendation. We have edited all the suggested sections in the revised manuscript. Kindly refer to the mentioned sections of the revised document.

6. There is sentence error in some section of this paper.

Again, thank you so much for picking up proofreading issues. The revised manuscript has been proofread by an independent, expert review.

Reviewer 2

• Although the justification for two separate COSs (for intermittent claudication and CLTI) is clear, the potential for overlap between the two outcome sets (e.g., QoL measures) should be discussed in more depth.

Thank you so much for the feedback. Indeed, there is a good chance for a degree of overlap between the core outcome sets especially when it comes to Patient-reported Outcome Measures, PROMs. Nevertheless, specific outcomes (e.g., QoL tools) can be recommended for each core outcome set as there are certain QoL tools and PROMs validated for Intermittent claudication but not for CLTI and vice versa as well. Further clarification has been provided in the revised manuscript.

• While the Delphi process is well described, it is unclear how "borderline" outcomes (those close to the 70% threshold) will be handled.

Thank you so much for this valuable feedback. The borderline outcomes (i.e., non-consensus items) either close to the threshold or not will be carried over to the subsequent round. If by the end of the final round of the survey, there remains outcomes without consensus, they will be carried over for discussion in the final stakeholder meeting for either inclusion or exclusion. Further clarification has been added to the revised manuscript.

• The target of 100 participants for the Delphi survey is appropriate, but greater emphasis on ensuring diversity across geographical regions (especially since PAD presentation can vary) would strengthen the plan.

Thank you so much for the critique. The target number is rather arbitrary and we are hoping for far more participants and you have rightly recommended ensuring diversity is far more important that is why we are collaborating with a very wide professionals and patients across the globe. Members of the stakeholders represent Europe, North America, Japan, Australia and New Zealand, South East Asia, and Middle East. All which will disseminate the Delphi surveys within their respective networks. Utmost attempts will be made ensure the participants are as diverse as possible.

• While the data availability statement is compliant, given that this is a COS development, transparency about access to the list of outcomes generated (even preliminary) would be important.

All the generated data; raw, preliminary, and final data sets will be made available starting from the search strategies, reported outcomes lists from systematic reviews and interviews, etc. A separate section further clarifying the availability of data has been added to the revised manuscript.

• Occasional minor typos, a final proofread before acceptance is recommended.

thank you so much for picking up proofreading issues. The revised manuscript has been proofread by an independent, expert review.

• Figure 1 (Key Steps of the Study) is mentioned but not included in this file.

Apologies for the mistake. All the included tables and figures will be uploaded separately as well.

• The search strategies are well presented; however, the timeframes for searches (especially if new studies are published during the study period) should be clarified.

As the generation of the outcome lists is a reiterative process, the search strategies for the systematic reviews will be continuously updated to include new studies for potential unique outcomes until the point of generating the outcome lists from the results of the systematic reviews and the qualitative interviews. This process has been further clarified in the revised manuscript.

---

## [Decision Letter · Decision Letter 1]

Core Outcome Sets in Symptomatic Peripheral Artery Disease, COS-PAD: Study Protocol for Developing Core Outcome Sets in symptomatic PAD utilising Systematic Reviews, Interviews, and Delphi Consensus.

PONE-D-25-06016R1

Dear Dr. Shwan,

We’re pleased to inform you that your manuscript has been judged scientifically suitable for publication and will be formally accepted for publication once it meets all outstanding technical requirements.

Kind regards,

Biswabandhu Jana, Phd

Academic Editor

PLOS ONE

Additional Editor Comments (optional):

Reviewers' comments:

Reviewer's Responses to Questions

**Comments to the Author**

1. Does the manuscript provide a valid rationale for the proposed study, with clearly identified and justified research questions?

Reviewer #1: Yes

2. Is the protocol technically sound and planned in a manner that will lead to a meaningful outcome and allow testing the stated hypotheses?

Reviewer #1: Partly

3. Is the methodology feasible and described in sufficient detail to allow the work to be replicable?

Reviewer #1: Yes

4. Have the authors described where all data underlying the findings will be made available when the study is complete?

Reviewer #1: No

5. Is the manuscript presented in an intelligible fashion and written in standard English?

Reviewer #1: Yes

You may also provide optional suggestions and comments to authors that they might find helpful in planning their study.

Reviewer #1: All typographical errors present in the document should be identified and corrected to ensure clarity and professionalism. Additionally, any highlighting or use of yellow color—whether for emphasis or tracking changes—should be removed to maintain a clean and consistent formatting throughout the text

**Do you want your identity to be public for this peer review?** For information about this choice, including consent withdrawal, please see our Privacy Policy

Reviewer #1: No

---

## [Editor Report · Acceptance letter]

PONE-D-25-06016R1

PLOS ONE

Dear Dr. Shwan,

I'm pleased to inform you that your manuscript has been deemed suitable for publication in PLOS ONE. Congratulations! Your manuscript is now being handed over to our production team.

Kind regards,

on behalf of

Dr. Biswabandhu Jana

Academic Editor

PLOS ONE